# The Application of Hyaluronic Acid Injections in Functional and Aesthetic Andrology: A Narrative Review

**DOI:** 10.3390/gels9020118

**Published:** 2023-02-01

**Authors:** Nicolò Schifano, Paolo Capogrosso, Gabriele Antonini, Sara Baldini, Fabrizio Scroppo, Andrea Salonia, Nicola Zerbinati, Federico Dehò

**Affiliations:** 1ASST Sette Laghi–Circolo e Fondazione Macchi Hospital, 21100 Varese, Italy; 2Antonini Urology, 00185 Rome, Italy; 3Department of Urology, Università Vita-Salute San Raffaele, 20132 Milan, Italy; 4Unit of Urology, Division of Experimental Oncology, Urology Research Institute (URI), IRCCS Ospedale San Raffaele, 20132 Milan, Italy

**Keywords:** hyaluronic acid, penile girth enlargement, Peyronie’s disease, premature ejaculation

## Abstract

Hyaluronic acid (HA) is a glycosaminoglycan widely utilised in different fields of medicine. We aimed to provide a comprehensive overview of the scientific evidence on the use of HA in andrology. A review of the literature to identify pertinent studies concerning the use of HA in andrology was carried out on the Medline, EMBASE, and the Cochrane databases, with no time restriction up to December 2022. Penile girth enlargement (PGE) using HA proved to be safe and effective in enhancing the diameter of the penis, with durable and satisfactory outcomes in long-term follow-up. Injection of HA in the glans seems to represent an alternative treatment option for those patients with premature ejaculation (PE) who fail to respond to conventional medications. HA intra-plaque injections represent a valid option which may contribute to restore sexual activity in patients with Peyronie’s disease (PD). The adoption of HA filler injections should always be tailored to the patient’s peculiar anatomy and underlying condition. More robust evidence is required to achieve a uniformed consensus regarding the use of HA in andrology, and further efforts should continue to improve the current injection techniques and HA products.

## 1. Introduction

The appearance of the penis and the males’ sexual performance are intimately connected with the self-confidence of men in modern society [1], and it is not surprising that patients seeking medical help for penile girth enhancement (PGE), premature ejaculation (PE) and Peyronie’s disease (PD) are on the rise more recently. In this era of growing emphasis on minimally invasive procedures, the popularity of soft tissue fillers has grown, with a renewed interest in their andrological applications.

Penile size concerns cause even patients with average-sized penises to seek penile augmentation [1], even though this field remains a controversial topic in reconstructive andrology. A variety of surgical and non-surgical strategies have been developed to address the increasing demand from these patients. Even though there is initial evidence [2] that levels of penile lengthening can be obtained through non-invasive strategies, effective enhancement of the length of the penis is mainly obtained through invasive surgery. Meanwhile, the demand for procedures to enlarge genital girth has increased significantly, even though a major concern is that penile girth enhancement is primarily performed for aesthetic purposes, unlike penile lengthening [1]. Although fillers have been comprehensively investigated for their use in other areas of the body, their applications in the penis pose different challenges due to the different anatomy and the larger amount of filler needed for the penis, thus requiring dedicated clinical studies. A number of different fillers have been examined for use in PGE [1], even though high-quality evidence on their efficacy and safety is scant, and there is still limited guidance on their use in clinical practice.

Premature ejaculation (PE) is a highly prevalent male sexual disorder, with a prevalence ranging from 8% to 30% of males [3]. Several molecules have been used in its treatment [4], including local anaesthetics, long- or short half-life SSRIs (e.g., dapoxetine), and opiates. Glans penile augmentation (GPA) using fillers has been proposed as an alternative option for PE patients who are resistant, or for those who experience significant side effects with the pharmacotherapy [5].

Peyronie’s disease (PD) is a highly prevalent fibrotic disease characterised by the deposition of collagen plaques in the tunica albuginea of the penis, leading to penile deformity, pain, erectile dysfunction (ED), and eventually a detrimental impact on the patients’ quality of life. The conservative management of PD is primarily focused on patients in the early (i.e., acute) stage, whilst surgery is typically reserved for patients in the stable phase of the disease [6]. Among the several options suggested for non-surgical PD treatment, injections of pharmacologically-active compounds directly into the PD plaques still represents the most popular treatment modality, as current evidence does not support the use of oral agents. 

Hyaluronic acid (HA) has been identified as the most ideal filler in a number of clinical specialties, thus including andrology [7]. HA is the predominant glycosaminoglycan in the extracellular matrix, consisting of glucuronic acid and N-acetylglucosamine held together by β-glycosidic bonds [7]. HA stabilizes intercellular connections through chemical bonds with collagen fibres, contributes to the cells’ proliferation and migration processes, and may also induce neocollagenesis by influencing the structure and function of the extracellular matrix. The biochemical structure of HA is similar across all species and its potential for immunological reaction is therefore negligible [8]. Biochemical reticulation (e.g., cross-linking) stabilizes the HA molecule such that it resists degradation by hyaluronidases, thus improving its longevity without a decrease in biocompatibility [9]. In a physiological condition, the hyaluronate molecule is highly polar and water-soluble. The isovolemic degradation constantly maintains the gel in balance with the water, thus maintaining the effect even in low concentrations of the filler [10]. Overall, the effect of HA fillers is usually long-lasting depending upon several characteristics, including the cross-linking levels, the purity and the concentration of HA in the filler material. 

Although injective HA treatment in andrology is increasing, current criticism of its andrological applications is secondary to the lack of established procedures, poorly defined indications, and concerns with the reliability of the existent scientific literature in this field. In this paper, we performed a comprehensive narrative review of the literature on the use of HA in andrology in order to guide the clinician to responsibly counsel those patients seeking medical help for small penis syndrome, PE and PD.

## 2. Evidence Synthesis

### 2.1. Penile Enlargement

Penile size has gained importance over time, with adequate dimensions corresponding to a perception of an advantage from both an aesthetic and a sexual standpoint. Managing penile-dimension related concerns may represent a significant challenge for the andrologist, especially when the patients suffer from penile dysmorphophobic disorder (PDD). A multidisciplinary evaluation, including a psychiatric and/or psychological assessment, may help in discerning those patients who would benefit the most from counselling from those who would benefit from interventional management instead [1]. The treatments being available to address the subjective concerns of those patients seeking medical advice regarding the size of their genitals have been criticised due to a number of limitations, such as possible complications, suboptimal aesthetic outcomes, and most crucially patients’ unrealistic expectations. The management approach should always be tailored to the subject’s unique anatomy and underlying conditions in these patients, and minimally-invasive procedures should always be preferred in this context. Largely used as soft tissue filler for aesthetic purposes, HA has also been successfully adopted as a girth-enhancement injectable compound for penile-shaft diameter augmentation [11]. HA can also be used to obtain enhancement of the dimensions of the glans [12] (Figure 1), which may have both an aesthetic and a functional usefulness, as a small glans with a thicker shaft may lead to penetrative issues. The conical shape of the glans allows for easy intromission of the penis into the vagina [13]. Patients usually desire that their penis appear cosmetically normal, and appropriately sized as compared to the penile shaft.

Kim et al. [5] injected HA at the proximal one-third of the penis from the tip of the glans to the coronal sulcus in a series of 187 patients suffering from low self-esteem due to perceived small penis. One year later, the net increase in maximal glandular circumference was 14.93 ± 0.80 mm in the subset of 100 naive patients and of 14.78 ± 0.89 mm in 87 patients who received a previous unsatisfactory dermo-fat graft. One year after the injection, 95% of Group 1 and 100% of Group 2 maintained more than 50% of the injected volume, based on the subjective patient’s visual estimation. In addition, the proportion of postoperative satisfaction as measured by the visual analogue scale (VAS) was 77% for Group 1 and 69% for Group 2. Kwak et al. [14] evaluated 50 patients with small penis syndrome injected with HA fillers. Compared to a baseline circumference of 7.48 ± 0.35 cm, the maximal penile circumference did increase significantly up to 11.41 ± 0.34 cm at 1 month (*p* < 0.0001), with these results remaining substantially unchanged after 18 months (11.26 ± 0.33 cm). The VAS at the latest follow-up evaluation confirmed the patients’ and partners’ stated levels of satisfaction throughout the follow-up. Micheels et al. [12] reintroduced the “Mushroom technique” proposed by Sito, which aimed to boost sexual feeling in both the treated individuals and their partners. In 12 patients, high-density HA gel was injected circumferentially around the corona and on the surface of the glans. Participants were given a multiple-choice self-assessment questionnaire, but the glans diameter was not objectively assessed by a clinician. All of the treated individuals observed an increase in the sexual sensation and enlargement of the glans. In a retrospective study including 83 patients, Sito et al. [15] compared the outcomes of HA emicircumferential penile shaft injections with those obtained using a similar technique with lipofilling. The increase in penile flaccid circumference obtained with both of the procedures ranged from 3.2 to 4.5 cm, and more than 80% of patients were “highly satisfied” with the outcomes. In the lipofilling arm, the safety profile was more unfavourable, with granuloma being found in 7/27 patients and fat necrosis with skin loss occurring in 1 of 27 patients. The operative time was also longer in the lipofilling arm. Yang et al. [11] published their results from a randomised, multicentre, patient/evaluator-blinded trial evaluating the outcomes with HA fillers and polylactic acid (PLA) injections for PGE in 72 patients seeking medical advice for PGE, with a 48-week follow-up. Both groups experienced a significant and sustained augmentation of the penile girth (16.95 ± 10.53 and 13.49 ± 9.98 mm of mean increase in the HA and PLA groups, respectively; *p* < 0.001). Interestingly enough, after 4 weeks of therapy, HA considerably outperformed PLA (*p* < 0.001), although after 48 weeks, no differences were noted between the two groups. Similarly, the degree of satisfaction with penile appearance as measured by a VAS score increased after the procedure, an effect which was maintained without significant differences between the two groups at the conclusion of the follow-up period. In the subsequent investigations [16,17], the same Authors largely validated similar results. Ahn et al. [18] recruited 64 participants in a prospective, randomised, controlled multicentre trial, comparing a HA filler to a PLA filler. The mean increase in penile girth was 22.74 ± 12.60 mm and 20.23 ± 8.73 mm in the HA and control groups, respectively (*p* > 0.05). Satisfaction levels concerning penile cosmesis and the quality of the sexual life significantly improved in both groups. The IELT also significantly improved in the HA group (e.g., from 5.36 ± 3.51 to 7.86 ± 4.73 min, *p* = 0.0001) and control group (e.g., from 5.23 ± 3.55 to 6.43 ± 4.22 min, *p* = 0.021). Although the exact reason for improvement of the IELT has not been conclusively established, it is believed to be similar to that of glans penis HA filler injections for treating PE. It is hypothesised that the filler injected between Buck’s fascia and dartos fascia acts as a barrier between tactile inputs and the dorsal-nerve ending receptors in the penile shaft, hence lowering the sensitivity threshold. The possible influence on ejaculation does not seem to have any detrimental effect on sexual life satisfaction. Quan et al. [19] identified 230 patients who received HA injections for penile augmentation. The penile circumference rose by 2.66 ± 1.24 cm, 2.28 ± 1.02 cm, and 1.80 ± 0.83 cm, respectively, during the 1-month, 3-month, and 6-month postoperative follow-ups. Zhang et al. [20] identified thirty-eight patients who underwent PGE using HA injections. Compared to baseline measurements, flaccid penile circumference and length significantly increased by 3.41 ± 0.95 cm (*p* < 0.01) and 2.55 ± 0.55 cm (*p* < 0.01) at the first month post-injection. At 12 months, despite attenuations, statistically significant improvements in flaccid penis size were still obtained, namely 2.44 ± 1.14 cm in girth (*p* < 0.01) and 1.65 ± 0.59 cm in length (*p* < 0.01). An overview of the studies dealing with the adoption of HA injections to obtain PGE is available in Table 1.

**Figure 1 gels-09-00118-f001:**
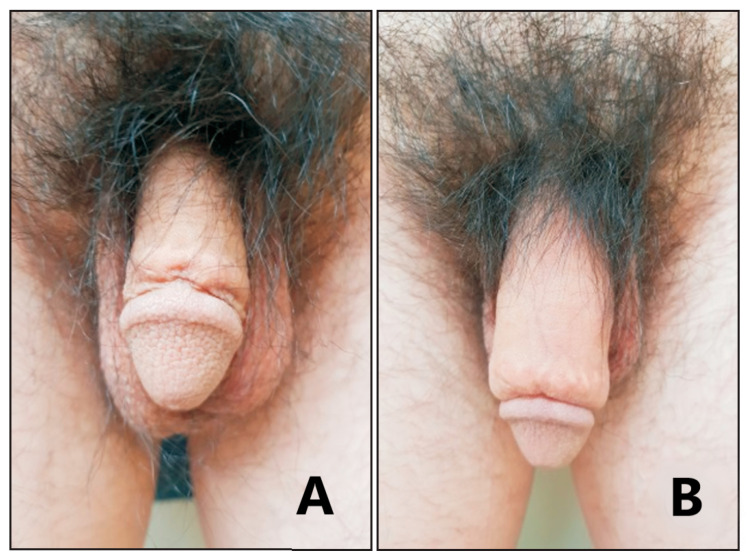
Penile girth enhancement using a HA filler. Penile appearance before injection (**A**) and after injection (**B**). Images reused from Ahn et al. [18], which is distributed under the terms of the Creative Commons Attribution Non-Commercial License (http://creativecommons.org/licenses/by-nc/4.0, accessed on 25 January 2023).

There is likely an underestimation of the complications from PGE. Though the adverse effects of the HA injection are usually mild and rare, these may still cause significant dissatisfaction in the affected patients. Ahn et al. [21] performed a systematic review of the literature, aiming to describe the possible risks of the use of PGE using the HA fillers. HA fillers were found to potentially cause adverse events early after the procedure or even years later. The most common adverse outcomes were cosmetic dissatisfaction, overzealous correction, irregularities of the penile surface, formation of granulomas, and possible necrotic changes secondary to vascular impairment. In the study by Quan et al. [19], during the entire 6-month follow-up, 4.3% of the patients had complications, such as subcutaneous bleeding, subcutaneous nodules, and infection, whilst penile oedema was found in 21/230 patients, all of which were uncircumcised before the procedure. The Authors identified a possible causal relationship between postoperative penile oedema and preoperative redundant prepuce, as this was hypothesised to cause distal accumulation of the injected HA gel, leading to lymphatic compression. Interestingly enough, in the RCT by Abdallah et al. [22], where all the patients were circumcised prior to HA gel injection, no penile oedema was reported. Dealing with remaining penile deformities following surgery or after injections of nonabsorbable substances is a troublesome issue. Even though the evidence in this respect is scant, the HA fillers can also be used with satisfactory and reproducible outcomes to correct residual penile girth deformities after PGE surgery. In the study of Kim et al. [5], the HA injections were performed via cannulation with 27G and 30G needles in the glans, which allowed for spontaneous closure of the needle entrance sites without suturing. Using 18G cannula has been reported to ease the distribution of the HA filler, but the injection sites might not heal spontaneously, and suturing of the injection site may be needed in order to prevent possible contamination of the implant with the genital bacterial flora [23]. Infection is a rare outcome after HA injection for PGE purposes, but can lead to detrimental outcomes. In a case report [23], a 35-year-old patient developed infection of the penis with gross oedema and erythema. Within 24 h, the patient developed septic shock with anuria, requiring transfer to the Intensive Care Unit (ICU). One month post admission, there was significant superficial skin loss to both the ventral and lateral parts of the penis.

### 2.2. Peyronie’s Disease (PD)

The aetiology of PD remains to be better elucidated, but it was proposed that repeated micro-traumatisms during sexual activity could represent the initiating factor, leading to a prolonged inflammatory reaction to the tunica albuginea of the penis in predisposed patients [24]. PD typically causes ED and penile deformity, which can impair penetrative sexual activity, thus significantly impacting the quality of life of the affected individuals. Non-surgical treatments, which can be proposed to the affected patients, are reserved for the acute phase of PD with the aim of attenuating the course of the disease, improving sexual function and managing pain [6]. Surgical treatment is usually reserved for the chronic stage of the disease [6], even though penile prosthesis implantation can lead to favourable results even before the disease evolves into the chronic phase [25]. Currently, different non-surgical management options are available to treat the acute phase of PD. These options include oral medications (e.g., potassium para-aminobenzoate (Potaba), vitamin E, colchicine, phosphodiesterase type 5 inhibitors (PDE5Is)); intralesional injections (e.g., steroids, verapamil, collagenase clostridium histolyticum (CCh), interferons); and local treatments (e.g., low-intensity extracorporeal shock wave therapy (LiESWT), traction and vacuum devices) [6]. Notwithstanding the wide range of available treatments, only Potaba and CCh (i.e., which is currently not available in the European market) have been approved by the FDA. Therefore, new options to treat patients in the acute phase of PD are needed. In this context, the use of intralesional HA injections has become common more recently due to its anti-inflammatory effects. HA is in fact capable of decreasing oxygen-free radicals production, modulating cell apoptosis, and decreasing scar formation [26], all effects which can potentially be useful in treating the acute phase of PD.

In a prospective, single-arm multicentric series, Zucchi et al. [27] showed that intraplaque HA injections significantly reduced both plaque size and penile curvature and improved the IIEF in 65 patients with PD, although the effects on penile curvature were inconspicuous (e.g., median decrease of 10 degrees in the curvature extent). The study was mainly limited by the lack of a control group and the short follow-up (namely 3 months). Gennaro et al. [28] reported comparable results in a further study comprising 83 patients treated with intralesional HA vs. 81 men in the control group with a 24-month follow-up. Similarly to what was previously seen [27], curvature improvement levels were statistically significant after treatment, although the treatment did not dramatically change the curvature extent (e.g., −9.0°, *p* < 0.0001). In a recent multicentric, prospective, randomised trial, Favilla et al. [29] found a small (e.g., −4.9°) but still statistically significant influence of HA injections on penile curvature, and a greater efficacy of HA in terms of patient satisfaction compared to verapamil injections. Their trial suffered, however, by some possible issues, mainly secondary to the absence of a placebo control group, and because of the short follow-up (e.g., 3 months). Cocci et al. [30] identified similar findings in a prospective, non-randomised trial comparing two groups of PD patients treated with a course of HA and verapamil infiltrations, respectively. Those patients treated with HA injections demonstrated a substantial reduction in the plaque-size and degree of the recurvatum (e.g., −9.5°), along with a significant IIEF-15 improvement compared to those treated with Verapamil. Cai et al. [31] published a prospective, randomised phase III study with 81 PD patients enrolled at two centres and randomised to oral HA administration combined with intralesional HA vs. intralesional treatment only. The combination of oral and intraplaque HA was found to promote better results regarding curvature improvement and overall sexual satisfaction (*p* < 0.001) vs. intralesional HA only. An overview of the studies dealing with the adoption of HA injections to treat PD is available in Table 2.

The majority of the above-mentioned trials were relatively large, had a prospective randomised design, and all of them consistently showed a reduction in the volume of plaque, along with a clinically modest but statistically significant decrease in the severity of the curvature. In conclusion, HA is emerging as a valid choice for the treatment of acute-phase PD, may contribute to the stabilisation of the disease and to improvement in the clinical picture, and should be regarded as a viable early option before considering any other possibilities.

### 2.3. Premature Ejaculation (PE)

PE is defined as an ejaculation that nearly always occurs before or within about 1 min of vaginal penetration. Penile sensitivity is the result of multiple interfering factors, including the dorsal nerve distributions, the number of receptors, their sensitivity-threshold, and the accessibility of the tactile stimuli to these receptors. To date, a number of treatment modalities have been implemented for PE, including topical anaesthetic agents, selective serotonin reuptake inhibitors (SSRIs), and opiates [4]. Even if all of these treatments have shown to be highly effective, the discontinuation rates are still substantial, mainly due to side effects such as anorgasmia, ED and decreased libido for the SSRIs and decrease in sexual pleasure in both of the partners for topical anaesthetics [32]. Surgical treatment (e.g., dorsal penile neurectomy) can represent an alternative treatment option [33], but its adoption is typically discouraged due to the non-negligible risk of permanent loss of penile sensitivity. A bulking agent such as HA being injected into the glans penis just above the nerve terminals is capable of inhibiting the tactile stimuli to reach the nerve receptors. The human glans is covered by stratified squamous epithelium and a dense layer of connective tissue where the nerve endings are present. When glans erection happens, this elongates the underlying elastic rete ridge, thus exposing more underlying sensory receptors [13]. 

Kim et al. [5] reported the outcomes of 139 PE patients being treated with either dorsal neurectomy (Group 1); dorsal neurectomy + HA glans augmentation (Group 2); or glans augmentation with HA only (Group 3). All groups demonstrated a considerable increase in IELT (up to 2.9 folds) as compared to their baseline scores. The same authors [14] reported the results of a 5-year follow-up in 38 patients, documenting a decrease in the IELT compared with the 6-month assessment, but this was still longer compared to baseline (e.g., 4.2 folds). Moreover, they reported that the treated patients had a satisfactory sexual life in 76% of cases, even at the 5-year follow-up. Abdallah et al. [22] examined the outcomes of HA glans injections in 60 PE patients in a randomised non-controlled trial. They compared the effects of two different HA-injection techniques, namely the “fan” (Figure 2) and the “multiple puncture” techniques. With the “fan technique” the injection needle was pushed at the proximal one-third of the glans and to the coronal sulcus from the distal part of the glans, and was angulated to both sides to inject the material as evenly as possible. The “multiple puncture technique” allows for a more uniform distribution of the injected material throughout the tissue. This strategy is based on injecting small amounts of HA through multiple points of entry, starting from the proximal one-third of the glans and coronal sulcus and proceeding to the distal tip. At the 1-month follow-up, the IELT increased in both groups (e.g., 3.6 folds, 2.12 ± 1.16 to 7.71 ± 7.86 min), without any significant difference between the two groups. The multiple puncture technique was associated with less pain and discomfort than the fan technique because the HA bullae formed with the multiple puncture technique were smaller. However, disadvantages of the multiple puncture technique included longer injection times and a higher risk of bruising. Littara et al. [34] identified 110 PE patients who were treated with multiple HA injections in the glans. The surface of the glans was subdivided into three circles from the base of the glans, at about 1-cm distance from each other. The circles were then further sub-divided into quadrants. An injection containing 1 mL HA was performed in the deep dermis into every quarter circle for a total of 12 injections, showing a 3.3-fold increase in IELT at 6 months post-treatment, from 88.34 ± 3.14 to 293.14 ± 8.16 s. Alahwany et al. [35] performed a placebo-controlled study randomising 30 PE patients to receive either HA glans injection or saline injection. The prefilled syringes were injected with a 30 G needle using the multiple puncture technique at two circular levels: one at the level of the corona and the second one mid-way between the corona and the urethral meatus. Six injections were performed at coronal level, and four in the second level. Significant IELT improvement after 1 month was identified in the treatment group, with the magnitude of increase being in the range of 2.6-fold higher vs. baseline compared to the 1.1-fold increase for patients receiving saline (*p* = 0.001). Shebl et al. [32] aimed to demonstrate the safety and efficacy of HA injections in the glans penis for the treatment of PE. Forty patients (group A) underwent HA injection by a four-inlet injection technique, and the same number of patients (group B) underwent saline injection in the glans by using the same technique. At the end of the six-month follow-up, the IELT significantly improved (e.g., 4.5 folds) in the HA injection group, compared to the baseline values and the control group. The maximal glandular circumference significantly increased, and the rate of patient satisfaction was 64.9%, 70.3% and 78.4% at the 1st, 3rd and 6th month of follow-up, respectively. An overview of the studies dealing with the adoption of HA injections to treat PE is available in Table 3.

Adverse reactions were minimal and merely self-limiting. The main adverse events were local discomfort, ecchymosis and papule formation, all of which were reported to resolve spontaneously. Initial discoloration by glandular swelling was frequently reported, and especially so when injections were too superficial, but this typically recovered to normal within 2 weeks [21]. Minor surface undulation originating from the undulation of the underlying rete ridge may alter the natural appearance of the penis, but it disappeared during erection and most patients were still satisfied. Preoperative circumcision was reported to provide a potentially better aesthetic result in those patients undergoing this procedure [19], and may have an independent role by itself in decreasing penile sensitivity and eventually the IELT. Although a significant increase in the IELT after this procedure is to be considered as the desired outcome in patients with PE, the majority of these studies identified levels of decreased penile sensation persisting for a long time after the procedure itself. Although no serious adverse reaction was reported, it was hypothesised that overzealous glans augmentation using HA fillers and/or too deep injections were potentially associated with deep vascular compromise [36].

### 2.4. Limitations

A narrative-design was chosen for this review to allow for broader coverage of the literature and for more flexibility. We recognize that this choice has imposed some possible limitations. The criteria for article selection were not made explicit for several of the selected studies, potentially leading to selection bias. To address this issue, a thorough set of original papers has been chosen and cited, and a structured method has been utilized to identify the most relevant studies on the subject.

## 3. Conclusions and Future Directions

Overall, HA may represent an extremely well-tolerated compound allowing for effective application for treating male sexual health issues. Even though the application of HA injections in the everyday clinical practice of the andrologist is increasing, more high-quality scientific studies are still needed to confirm the efficacy and safety of these procedures. Even though the majority of currently available studies showed the efficacy and safety of HA injections in treating small penis syndrome, PD and PE, the use of different types of HA products with different dosages and different techniques prevent a reliable comparison of the results. More high-quality, randomised prospective studies and the standardisation of the inclusion criteria and the outcome assessment methods are needed in order to confirm these findings. The usefulness of self-assessment scales for patients with PD, PE, and penile size concerns is limited, due to the role of preoperative expectations in this category of patients. Although complications of HA injections are usually mild and rare, anecdotal reports of detrimental complications were reported, and these may affect the patients’ satisfaction after treatment. A careful patient selection is of paramount importance to increase postoperative satisfaction after HA injections in andrology, and mastering the injection technique by adequate training is necessary in order to achieve an optimal outcome.

## 4. Materials and Methods

A comprehensive search strategy with no time period restriction was carried out on the MEDLINE/PubMed database, EMBASE, and the Cochrane Libraries in November 2022 to identify the more recent relevant studies dealing with the use of HA in andrology. The search strategy used the following keywords in combination with both Medical Subject Headings terms and text words: “HA and andrology, HA and Peyronie’s disease (PD), HA and premature ejaculation (PE), and HA and penile enlargement.” Unpublished studies, studies without primary data (i.e., reviews, commentaries, and letters), and conference abstracts were excluded. Only articles published in English in peer-reviewed journals were selected. Studies were considered eligible only if they included >10 patients. Abstracts were reviewed by the panel for relevance to the defined review question. Two authors (NS, SB) independently scrutinised the titles and abstracts for relevance in order to identify those studies which needed a more thorough assessment through evaluating the full-text papers. A third author (PC) resolved the discrepancies in the selection process. The relevant studies were then selected and screened, and the data were analysed and summarised after an interactive peer-review process of the panel.

## Figures and Tables

**Figure 2 gels-09-00118-f002:**
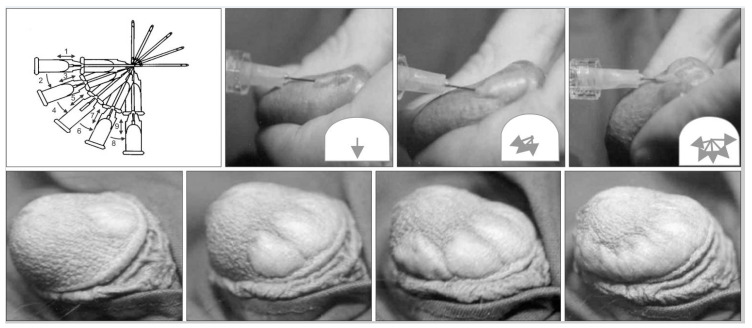
The “fan” technique. Images reused from Moon et al. [13], which is distributed under the terms of the Creative Commons Attribution Non-Commercial License (http://creativecommons.org/licenses/by-nc/4.0, accessed on 25 January 2023).

**Table 1 gels-09-00118-t001:** Studies dealing with the use of HA for penile enlargement purposes.

Author	Patients	Type of Study	Treatment	Filler Injection-Related Adverse Events (%)	Follow-Up Duration (Months)	Penile Girth Enhancement (mm)
Yang [11]	36 + 36	RCT	HA vs. PLA	2.8	12	20.6 ± 10.9
Yang [16]	37 + 35	RCT	HA vs. PLA	5.1	6	21.0 ± 10.0
Yang [17]	33 + 34	RCT	HA vs. PLA	9.1	18	19.1 (13.5–24.7)
Sito [15]	56 + 27	Retrospective	HA vs. lipofilling	0	24	32.0–45.0
Kwak [14]	41	Clinical trial	HA	0	18	39.0 ± 0.3
Micheels [12]	12	Clinical trial	HA	0	12	Not objectively measured
Ahn [18]	32 + 32	RCT	HA vs. PLA	0	6	22.7 ± 12.6
Quan [19]	230	Prospective	HA	4.3	6	26.6 ± 12.4
Zhang [20]	38	Prospective	HA	7.9	12	24.4 ± 11.4

**Table 2 gels-09-00118-t002:** Studies dealing with the use of HA to treat PD.

Author	Patients	Type of Study	Treatment	Duration of the Treatment (Weeks)	Follow-Up Duration (Months)	Plaque Dimensions (mm)	Extent of the Curvature Correction (°)	IIEF Score	VAS Score
Cai [31]	41 + 40	RCT	Intraplaque HA and oral HA	6	3	−3.0 ± 1.0	−7.8 ± 3.9	4 ± 0.3	−4.0 ± 2
Cocci [30]	125 + 119	Prospective	Intraplaque HA vs. verapamil	8	3	−1.50	−9.5	1.0	−4.0
Favilla [29]	69 + 63	Prospective	Intraplaque HA vs. verapamil	12	3	−1.80 ± 2.47	−4.6	1.78 ± 2.48	
Gennaro [28]	83 + 81	Case-control	Intraplaque HA	26	6−12−24		-9.0	3.8 ± 3.0	
Zucchi [27]	65	Prospective single arm	Intraplaque HA	10	2	−2.0	−10.0	1.0	−2.0

**Table 3 gels-09-00118-t003:** Studies dealing with the use of HA to treat PE.

Author	Patients	Type of Study	Treatment	Pre-Treatment IELT	Post-Treatment IELT
Littara [34]	110	Prospective	HA	88.3 ± 3.1	293.1 ± 8.1
Alahwany [35]	15 + 15	RCT	HA vs. saline	33.5 ± 14.8	73
Kim [5]	25 + 49 + 65	Prospective	HA	96.5 ± 52.3	281.9 ± 93.2
Kwak [14]	38	Retrospective	HA	84.2	376.7
Abdallah [22]	30 + 30	RCT	HA	132 ± 76	421
Shebl [32]	40 + 40	RCT	HA vs. saline	60	240

## Data Availability

No new data were created or analyzed in this study. Data sharing is not applicable to this article.

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
