# Peer review of "The Application of Hyaluronic Acid Injections in Functional and Aesthetic Andrology: A Narrative Review"

_gels, 2023, doi:10.3390/gels9020118_

Round 1

Reviewer 1 Report

This review is very well organized and meaningful to the use of HA in andrology. The authors aimed to provide a comprehensive overview of the scientific evidence on the use of HA in andrology, and they did what we wanted to do. I think the authors did very excellent jobs in summarizing the HA in andrology, however, it seems the author could add more information on the following:

1, the use of HA in glans augmentation: although most of the people may think the HA in premature ejaculation is the same as glans augmentation, well, they are not the same. The glans augmentation is more controversial as the “fine technique” injection method on the surface of the glans may eventually ended up some bulges on the glans surface, and how to yield to reasonable aesthetic result is the key. There are some literatures on glans augmentations, and I believe the authors could make a summary.

2, I would recommend the authors could add some illustrations on penile augmentation, premature ejaculation, etc.

Author Response

REVIEWER #1:

Q0 This review is very well organized and meaningful to the use of HA in andrology. The authors aimed to provide a comprehensive overview of the scientific evidence on the use of HA in andrology, and they did what we wanted to do. I think the authors did very excellent jobs in summarizing the HA in andrology.

A0 Many thanks for the positive feedback.

Q1 the use of HA in glans augmentation: although most of the people may think the HA in premature ejaculation is the same as glans augmentation, well, they are not the same. The glans augmentation is more controversial as the “fine technique” injection method on the surface of the glans may eventually ended up some bulges on the glans surface, and how to yield to reasonable aesthetic result is the key. There are some literatures on glans augmentations, and I believe the authors could make a summary.

A1: We would like to thank Reviewer #1 for these very insightful suggestions. Indeed HA injections in the glans penis can be used both to treat premature ejaculation patients and to address the concerns of those patients who seek medical advice for penile enlargement procedures. These very different categories of patients require indeed a particular and dedicated technique aiming at obtaining the final goal of these procedures, namely the patients’ satisfaction. Although achieving an optimal aesthetic outcome is desirable for both of these indications, cosmetic appearance remains particularly important for the penile enlargement patients. Whilst only two different HA glans injection techniques (e.g. the “fan” and the “multiple site” techniques) were described in the literature, few comparative studies between these different injection strategies were found in the review of the existing literature. Therefore, despite the “multiple site” technique probably carries the advantage of a better filler-distribution, with a possible better cosmetic result, the evidence is not strong enough yet to recommend conclusively on the use of one technique vs. the other in these very different categories of patients. Keeping in mind this premise, it is these Authors’ opinion that the injection technique, the number and the distribution of injection sites currently remains to be tailored according to each glans anatomical appearance and to the surgeons’ experience. These issues have been discussed as follows:

 “Abdallah et al[22] examined the outcomes of HA glans injections in 60 PE patients in a randomized non-controlled trial.  They compared the effects of two different HA-injection techniques, namely the “fan” and the “multiple puncture” techniques. With the “fan technique” the injection needle was pushed at the proximal one-third of the glans and to the coronal sulcus from the distal part of the glans, and was angulated to both sides to inject the material as evenly as possible. The “multiple puncture technique” allows for a more uniform distribution of the injected material throughout the tissue. This strategy is based on injecting small amounts of HA through multiple points of entry starting from the proximal one-third of the glans and coronal sulcus and proceeding to the distal tip. (…) The multiple puncture technique was associated with less pain/discomfort than the fan technique because the HA-bullae formed with the multiple puncture technique were smaller. However, disadvantages of the multiple puncture technique included longer injection times and higher risk of bruising”.

Q2: I would recommend the authors could add some illustrations on penile augmentation, premature ejaculation, etc.

A2: We would like to thank Reviewer #1 for this suggestion. A range of figures have been added to the manuscript

Reviewer 2 Report

Dear authors, 

An interesting review with an interesting subject.

Introduction - adapted to the subject

Material and Method - lines 408-423 I think that line 87 should insered before ,,Evidence synthesis,, 

There are few reviewers in the literature with this topic which makes this manuscript valuable 

The manuscript represents a conclusion of existing studies. I think that authors must take account the formatting errors of the text. 

Thank you

Author Response

REVIEWER #2:

Q0: An interesting review with an interesting subject. There are few reviewers in the literature with this topic which makes this manuscript valuable. The manuscript represents a conclusion of existing studies.

A0 Many thanks for the positive feedback

Q1 Material and Method - lines 408-423 I think that line 87 should insered before ,,Evidence synthesis,, 

A1: We would like to thank Reviewer #2 for pointing out this possible issue. Even though we recognize that the order of the sections of our manuscript may appear unusual, the Materials and Methods section was placed after the Conclusions according to the Author’s guidelines of this specific Journal.

Q2: I think that authors must take account the formatting errors of the text. 

A2: Many thanks. The text has been carefully and extensively checked for formatting errors, and corrections were made where appropriate.